# De novo protein design of photochemical reaction centers

Nathan M. Ennist [1,2,3] ✉, Zhenyu Zhao[1], Steven E. Stayrook [1,4,5], Bohdana M. Discher[1], P. Leslie Dutton[1] & Christopher C. Moser[1]

Natural photosynthetic protein complexes capture sunlight to power the energetic catalysis that supports life on Earth. Yet these natural protein structures carry an evolutionary legacy of complexity and fragility that encumbers protein reengineering efforts and obfuscates the underlying design rules for light-driven charge separation. De novo development of a simplified photosynthetic reaction center protein can clarify practical engineering principles needed to build new enzymes for efficient solar-to-fuel energy conversion. Here, we report the rational design, X-ray crystal structure, and electron transfer activity of a multi-cofactor protein that incorporates essential elements of photosynthetic reaction centers. This highly stable, modular artificial protein framework can be reconstituted in vitro with interchangeable redox centers for nanometer-scale photochemical charge separation. Transient absorption spectroscopy demonstrates Photosystem II-like tyrosine and metal cluster oxidation, and we measure charge separation lifetimes exceeding 100 ms, ideal for light-activated catalysis. This de novo-designed reaction center builds upon engineering guidelines established for charge separation in earlier synthetic photochemical triads and modified natural proteins, and it shows how synthetic biology may lead to a new generation of genetically encoded, light-powered catalysts for solar fuel production.

De novo construction of artificial photosynthetic reaction centers offers a means to test our understanding of biological electron transport and to re-engineer photosynthesis in ways that can be aimed directly at human needs. Natural reaction centers attain unidirectional, light-activated charge separation by manipulating electron transfer rates in chains of redox cofactors[1–3]. Linear electron flow in green plants and cyanobacteria begins with water as a source of electrons to produce $H_2$ equivalents for chemical energy, releasing $O_2$ in the process[4,5]. Oxygenic phototrophs span the wide redox range between water oxidation and proton reduction by using two reaction centers, photosystems I and II (PSI and PSII), to absorb two photons for each

electron abstracted from water[4,6,7]. In principle, a single visible photon absorbed by natural chlorophyll pigments is sufficiently energetic to power both reactions, suggesting an avenue toward increasing the overall solar-to-fuel energy conversion efficiency[4,8]. Our ultimate aim is to develop a one-reaction center photosynthetic system that supports water oxidation and proton reduction with minimal energy loss to heat.

Recent progress in de novo protein design has facilitated binding of multiple, varied small molecules and metal ions within electron tunneling distance of one another[9–15]. The next challenge for solar energy transduction in an artificial protein is to assemble a multi-step

[1]Department of Biochemistry and Biophysics, University of Pennsylvania, Philadelphia, PA 19104-6058, USA. [2]Institute for Protein Design, University of Washington, Seattle, WA 98195, USA. [3]Department of Biochemistry, University of Washington, Seattle, WA 98195, USA. [4]Department of Pharmacology, Yale University School of Medicine, New Haven, CT 06510, USA. [5]Yale Cancer Biology Institute, Yale University West Campus, West Haven, CT 06516, USA. ✉e-mail: ennist@uw.edu

electron transport chain that can convert photon energy to a charge-separated state that persists long enough to be used for chemical reactions such as fuel production. To this end, we have designed a photosynthetic reaction center protein maquette (the RC maquette) and solved its X-ray crystal structure in multiple states of assembly. The RC maquette achieves long-lived, light-activated charge separation and reproduces many of the elements of natural reaction centers: tyrosine and metal cluster oxidation reminiscent of the water oxidizing side of PSII, as well as reduction of low potential acceptors, including Co porphyrins known to participate in proton reduction to $H_2$[16,17].

## Results

### Engineering electron tunneling rates

Natural photosynthetic reaction centers achieve photochemical charge separation by anchoring a light-activated pigment between two electron transfer chains. One chain provides an electron acceptor for the excited pigment, and the other an electron donor that restores the pigment ground state while creating a charge-separated donor-acceptor pair (Fig. 1a). While electron accepting or donating chains have been developed in modified natural proteins[18–26] and in a de novo protein with a crosslinked ruthenium trisbipyridine derivative[15], our design condenses the extended electron transfer chains of natural reaction centers to a core elementary donor-pigment-acceptor triad, symbolized as DPA[27,28].

Upon light activation forming an excited pigment singlet state ($D^1P*A$), the pigment may undergo radiative decay to the ground state (DPA) or relax to the triplet excited state ($D^3P*A$) via intersystem crossing. This triplet state may in turn decay to the ground state by phosphorescence ($k_3$, Fig. 1b). During the excited state lifetime, electron transfer to a suitably placed acceptor forms $DP^+A^-$ with rate $k_1$. Charge-separated states can short-circuit through charge recombination ($k_4$, forming DPA) or undergo a further charge separation ($k_2$, forming $D^+PA^-$). In the absence of additional donors, acceptors or catalytic redox reactions the charge-separated state eventually decays by direct electron transfer ($k_5$) or indirect electron transfer through reverse reactions of $k_{-2}$ and $k_{-1}$ (Fig. 1b).

To engineer the appropriate cofactor spacing, we estimate rates of downhill intraprotein electron tunneling using a well-tested empirical expression, Eq. (1) (see Methods)[1,3,29]. This tunneling expression not only allows us to understand the engineering of natural electron transfer proteins[1,2,30] but also to consider the relative performance of artificial DPA triad designs in advance of construction (Fig. 1c, d).

### Light-active redox protein design

To begin reverse-engineering photosynthetic reaction centers, we use first principles of protein folding to construct a single-chain framework for positioning DPA cofactors that consists of an extended four α-helix

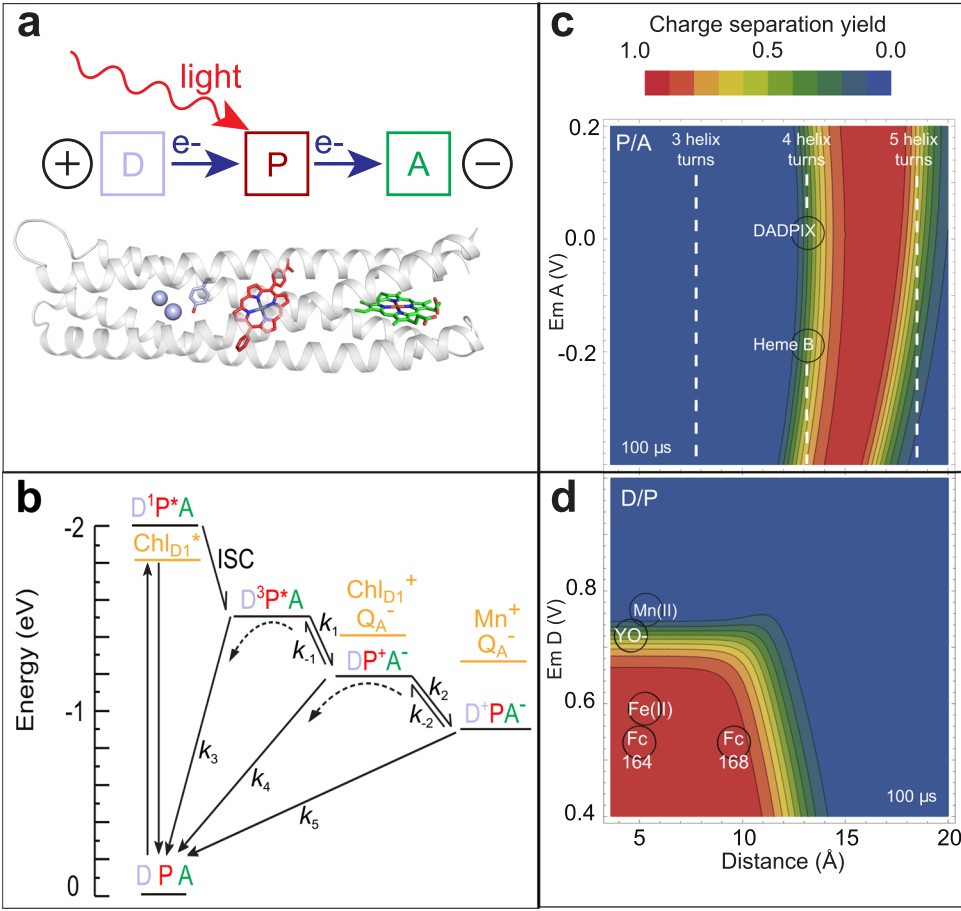

**Fig. 1 | Design of natural and artificial photosynthetic reaction centers. a** The light-activated donor-pigment-acceptor (DPA) electron transfer triad core of photosynthesis and X-ray crystal structure at 2.0 Å resolution of the RC maquette with a metal ion/tyrosine donor, Zn porphyrin pigment, and heme B acceptor (PDB ID: 5VJS). **b** Kinetic schemes of a light-activated system show energies plotted relative to ground state before light activation. An RC maquette DPA triad is compared with representative PSII charge separated states (orange)[74]. **c** Triad contour plots of expected relative charge separated $D^+PA^-$ yield after 100 μs for a range of P-to-A edge-to-edge distances vs. acceptor $E_m$ (using $E_m$ of 0.91 V for P/P[+] and $E_m$ of 0.72 V for tyrosinate donor). Dashed lines show cofactor anchoring residues adjusted in increments of -1 helical turn (-5.2 Å). Fe porphyrins heme B and DADPIX are shown as acceptor alternatives. **d** Corresponding triad contour plot for D-to-P distances and driving forces; acceptor is heme B; tyrosinate, Mn(II), Fe(II) and cysteine-coupled ferrocene (Fc164 and Fc168) shown as possible donors.

bundle coiled-coil. Such constructs are called maquettes[9,31–34] and they perform the same role as the scale maquettes made by architects to evaluate their designs prior to constructing the final product. Maquette amino acid sequences are not based on any specific natural protein. Instead, they employ charge complementarity and a binary pattern of polar and nonpolar residues to enforce a desired helical threading[9,32]. Further details of RC maquette design will be described elsewhere; see Supplementary Table 1 for amino acid sequences.

The RC maquette uses histidine (His) residues in the hydrophobic core to coordinate the central metals of tetrapyrroles for site-specific cofactor binding. A single His ligates a light-activatable Zn porphyrin in the pigment site[10–12,35], while two His residues on opposing helices bis-His ligate an Fe porphyrin in the acceptor site[10,14,31,36,37]. The RC maquette can bind different electron acceptors including heme B or Fe(III) 2,4-diacetyl deuteroporphyrin IX (DADPIX) tightly with $K_D \leq 10$ nM and with $E_m$ values of −0.19 and +0.01 V vs SHE, respectively. Amphiphilic Zn tetrapyrroles show the strongest binding to our de novo proteins[12]. In the results reported here, we use Zn 5-phenyl, 15-(p-carboxyphenyl) porphyrin (ZnP, $K_D \leq 10$ nM, $E_m$ 0.91 V) as our pigment, P. See Supplementary Fig. 1 for spectro-electrochemical determination of $E_m$ values of Fe and Zn porphyrins.

Distances between cofactors are the primary determinants of electron transfer rates in the RC maquette. Inter-cofactor distances are subtly influenced by core packing, the helical register, and the orientations of porphyrin rings, but larger distance changes are more easily accomplished by moving cofactor-ligating residues along the helices in increments of a full helical turn (-5.2 Å). Previous multi-cofactor de novo protein designs[10,13,14,38] have edge-to-edge distances between porphyrins that are too short or too long for high-yield charge separation in DPA triads as shown in the theoretical contour plot of Fig. 1c (calculated from Eq. 1 with the kinetic scheme of Fig. 1b). Instead, the RC maquette needs a -13–18 Å P-to-A distance (corresponding to four or five helical turns between the P- and A-site His residues) in order to achieve stable charge separation for various porphyrin acceptors with a wide range of redox properties. Details of the electron tunneling yield calculations are provided in Methods.

The contour plot of Fig. 1d indicates that the electron donor must be placed closer to P to assure sufficiently rapid electron tunneling rates that can compete with short-circuiting $DP^+A^-$ charge recombination, $k_4$ (see Fig. 1b). D-P distances up to -10 Å and $E_m$ values less than 0.75 V will give the best $D^+PA^-$ yield. To reproduce PSII-like photo-oxidation activity, we chose a Tyr-P distance of 5 Å for Tyr oxidation, and we assembled a metal cluster at a distance of 10 Å from P. The binuclear metal cluster uses a two-His/four-Glu motif seen in many natural metal clusters such as bacterioferritin[21]. The basis of our sequence selection in the metal cluster donor region of the RC maquette comes from an earlier de novo protein maquette named Due Ferri 3 (DF3)[39–41]. The sequence of DF3 was modified to integrate it into the electron donor region of the RC maquette in several ways. First, redox-inactive amino acids replaced Trp and Tyr residues that were not intended to be electron donors. Next, the loop from DF2t[42] replaced a His-containing loop from DF3 to avoid nonspecific His interactions with cofactors. A flexible linker was added to connect the two halves of the dimer. Finally, four helical Gly residues (one per helix) were inserted where the electron donor region meets the pigment binding site in order to increase local backbone flexibility and facilitate the transition from one domain to the next. The DeGrado and Lombardi groups independently developed an analogous strategy to fuse a DF protein to a synthetic Zn porphyrin-binding maquette[11]. The operation of the Tyr and metal ion donors in the RC maquette is compared with an unnatural maleimide-functionalized ferrocene donor (Fc) anchored to a Cys residue that replaces a Tyr residue at two different D-P distances (either Y168C for Fc168 or G164C/Y168L for Fc164).

Crystal structures of the RC maquette confirm the intended coiled-coil design with pigments and redox centers assembled at the designed positions (Fig. 2). To the best of our knowledge, this represents the first crystal structure of a designed protein containing two unique porphyrins.

The RC maquette has many structural features in common with natural proteins, despite the lack of sequence similarity and the absence of evolutionary natural selection. Hydrophobic core packing, rotameric states of heme-ligating histidines, and second shell histidine-threonine hydrogen bonds are similar to the cytochrome $b$ subunit of cytochromes $bc_1$ and $b_6f$ (Fig. 2a)[43]. Even the orientation of the maquette heme relative to the superhelical bundle axis is conspicuously similar, suggesting that the particular conformation of heme in the natural proteins is not precisely controlled for any functional purpose, but rather follows directly from structural constraints demanded by this protein fold. The light-active pigment binding site differs from the cytochrome $b$ heme site by providing only a single His for ligation.

In the electron donor module, a single Leu for His residue exchange (L71H mutation) in the bundle core immediately provides an H-bond for the Tyr hydroxyl, mimicking the proton-coupled electron transfer geometry of donor Tyr Z in PSII[44,45] (Fig. 2b). In the nearby metal binding site, Glu/His ligation of Cd(II) in the RC maquette accurately reproduces the geometry of the binuclear Mn(II) cluster shown to be photochemically active in modified bacterioferrin (Fig. 2c)[20,21].

Crystal structures with Zn(II) (PDB ID: 5VJS and 5VJT) show only one metal in the di-metal site, with a similar Tyr-to-closest metal distance as in the Cd(II) structure (PDB ID: 5VJU); the second metal site in the Zn(II)-containing structures is vacant (Supplementary Fig. 2). The Zn(II)-containing structures come from an acidic crystallization condition (pH 4.4–4.5) that likely protonates a metal site Glu, suppressing binding of the second metal. At the same time, nearby deprotonated Asp residues on the exterior are free to recruit a Zn(II) ion in a tetrahedral coordination geometry.

Spectroscopic titrations show that 2.0 pentacoordinate Co(II) ions bind the RC maquette at neutral pH in solution and that 2.0 Mn(II) ions bind the RC maquette-heme-ZnP complex with high affinity (Supplementary Fig. 3). The successful construction of the electron donor site in the L71H RC maquette represents a first step in exploring the Tyr-mediated photo-oxidative assembly of a PSII-like Mn cluster for water oxidation in a designed protein[46].

## Light-driven electron transfer dynamics

Transient absorption spectroscopy in the visible spectral region reveals charge-separated states and electron transfer rates in the RC maquette. Singular value decomposition (SVD) of the transient spectra (Fig. 3, Supplementary Figs. 4–11, and Supplementary Table 2) resolves a time and wavelength dependence that is fit to an elementary kinetic model of first order reactions between the excited, charge separated and ground states to generate rate constants (Fig. 3c) and spectra of the intermediate states (Fig. 3d). Details of data analysis are given in Methods.

When the RC maquette is assembled with only P and no donor or acceptor (Fig. 3b, ZnP monad and Supplementary Fig. 4a), a 2 ns 532 nm laser pulse bleaches the ground state Soret absorption of P at 424 nm, and the broad absorption from 440 to 550 nm of the excited triplet state increases, decaying to the ground state in 3–5 ms. This rate is similar to that observed in other artificial protein designs[12].

When an acceptor is also bound to make a PA (ZnP-ferric heme) dyad, the same triplet signature is observed but now decaying at $60 \pm 30$ μs, presumably because P* reduces the acceptor. The absence of any monad-like P* decay component with a 3–5 ms lifetime indicates a near-unity quantum yield of A reduction by P*. As expected, charge recombination in the dyad is rapid, because the large driving force closely matches the reorganization energy. Consequently, the $P^+A^-$ intermediate does not accumulate in the dyad, and a spectroscopic

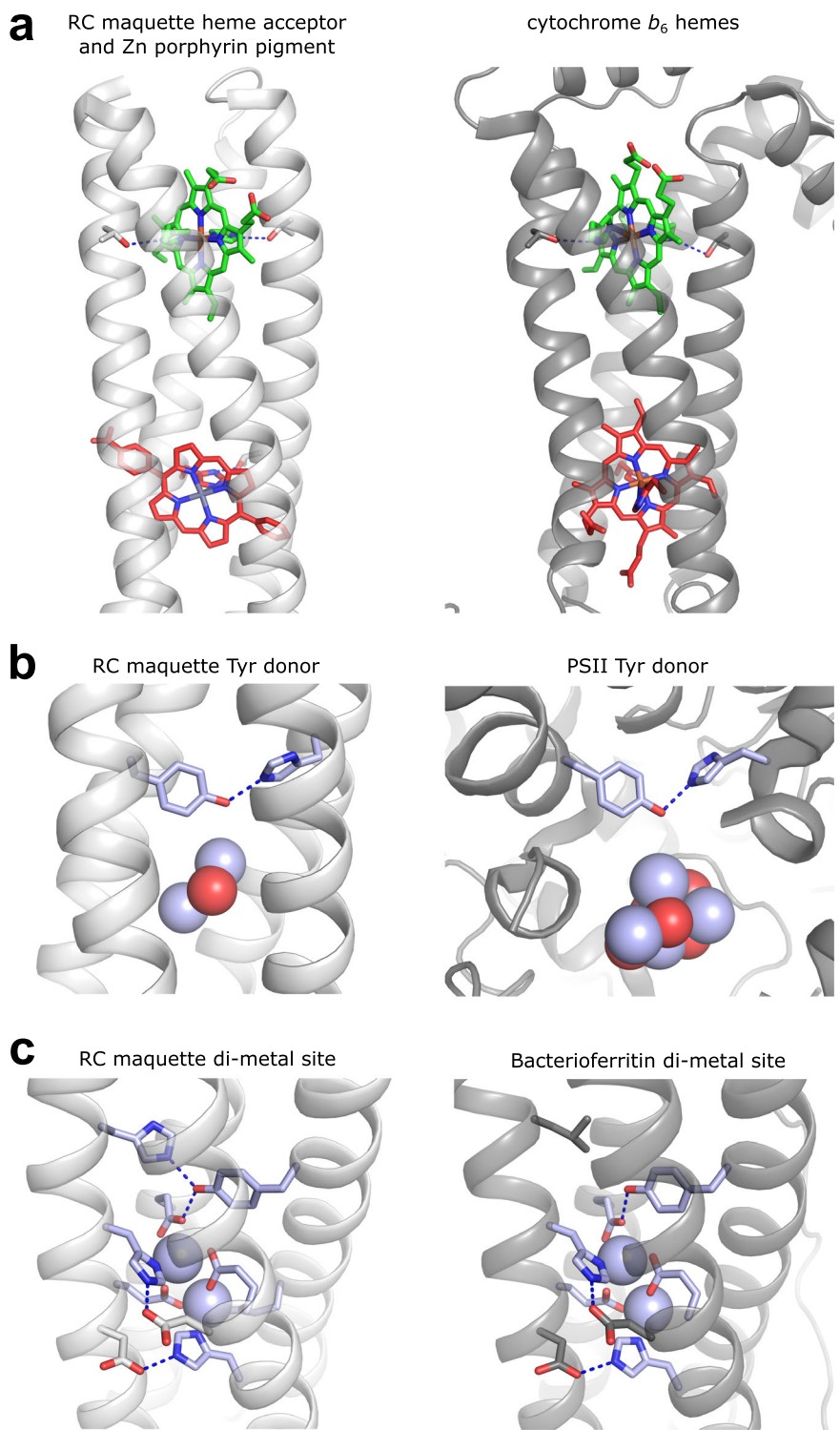

**Fig. 2 | Structural similarities between cofactor binding sites of designed RC maquette and natural proteins are significant despite lack of sequence identity.** RC maquette crystal structures are shown in white with heme electron acceptor and ligating His residues shown in green, ZnP pigment and His in red, and electron donating Tyr, metal ion spheres and first shell ligands in blue. Natural protein structures are shown in gray with cofactors colored as in RC maquette.

Metal-bridging oxygens are shown as red spheres. Blue dotted lines represent hydrogen bonds. **a** RC maquette (PDB ID: 5VJS) and cytochrome $b_6f$ (PDB ID: 6RQF[43]). **b** RC maquette-L71H mutant with Cd(II) (PDB ID: 5VJU) and PSII (PDB ID: 6DHE[44]). **c** RC maquette-L71H mutant with Cd(II) (PDB ID: 5VJU) and bacterioferritin with Cd(II) (PDB ID: 4CVS[21]).

signal of reduced heme cannot be resolved (see Fig. 3b and Supplementary Fig. 4b). Triplet inhibition of charge recombination is not apparent. When the heme acceptor in the dyad is inactivated by pre-reduction with dithionite, P* reverts to monad-like behavior with a 3−5 ms decay to the ground state.

To trap a long-lived charge separated state, the RC maquette requires adding a donor to the dyad to form a DPA triad. Substitution of donor Tyr for Leu168 stabilizes heme reduction in the $D^+PA^-$ charge separated state (see Y-ZnP-heme triad in Fig. 3a−d, Supplementary Figs 6−8, and Supplementary Table 3). The $D^+PA^-$ charge-separated

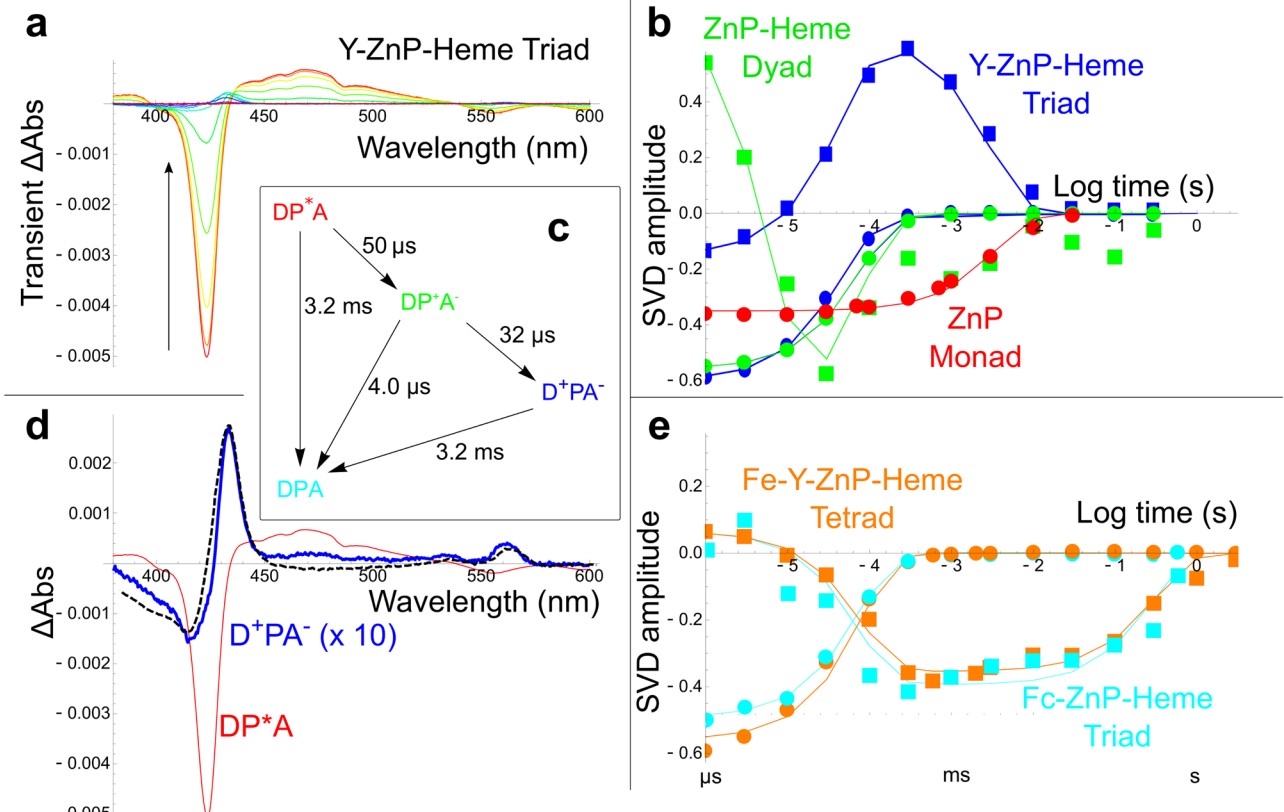

**Fig. 3 | Transient spectroscopy of RC maquette reveals the dynamics of light-activated charge separation and recombination. a** Difference spectra of the Tyr-ZnP-heme triad in L31D/L71H mutant at pH 9.5 are shown for delay times from 1 μs to 3 s after the laser flash at log time intervals shown in **b**. **b** The first and second principal time varying SVD components of the difference spectra (circles and squares, respectively) fit to a simple kinetic model connecting DP*A, DP*A⁻, D⁺PA⁻, and DPA states with single exponential first order reactions (solid lines): ZnP monad (red), ZnP-Heme dyad (green), Tyr-ZnP-heme triad (blue), (see also Supplementary

Figs. 4 and 6). **c** Fitted log rates for the Tyr-ZnP-heme triad. **d** Fitted DP*A and D⁺PA⁻ difference spectra for the Tyr-ZnP-heme triad dominated by ZnP bleach (red) and heme redox (blue) difference spectra, respectively, compared to a scaled heme redox spectrum acquired for a 20 μM RC maquette-heme sample reduced with dithionite in the dark (black dashed line). **e** Fitted time varying SVD components for the ferrocene-ZnP-Heme triad (cyan) and iron-Tyr-ZnP-heme tetrad (orange) show charge separation lifetimes up to 300 ms. Source data are provided as a Source Data file.

state difference spectrum identified by SVD (Fig. 3d, blue) is dominated by the signature of heme reduction (Fig. 3d, black dashed line) with minor contributions from ³ZnP* and ZnP⁺, providing conclusive evidence that heme functions as the electron acceptor in the Tyr-ZnP-heme B triad. The ³ZnP*/ZnP⁺ difference spectrum has very little shift in the Soret band[47], causing this difference to disappear in the noise in the Soret region. The heme redox signal persists for 3 ms, much longer than the 50 μs ³ZnP* bleach recovery. Known extinction coefficients for heme and ZnP provide a yield estimate of 11% from the excited triplet state, which in turn permits an estimate of Tyr to P⁺ ET of ~30 μs (Fig. 3c and Supplementary Fig. 6a).

As in natural PSII, Tyr oxidation is a proton-coupled electron transfer (PCET) reaction[45]. Depending on design and environmental conditions, electron transfer may involve Tyr deprotonation before thermodynamically favorable ET (observable at pH values near the Tyr p$K_a$ of ~10)[48], endergonic ET before PT, or concerted PTET. Figure 3a–d suggest tyrosinate ET at pH 9.5, with an $E_m$ value of ~0.72 V[48]. Engineered proton acceptors may increase the yield of electron transfer by lowering the p$K_a$ of Tyr and increasing the population of tyrosinate. While the introduction of a hydrogen bond from Tyr to a His residue appears to have little impact on the lifetime or yield of charge separation by itself, the addition of Asp31 adjacent to the His71-Tyr168 hydrogen bond raises the yield from 2 ± 1% to 11.5 ± 0.7% and shortens the lifetime of charge separation from 40 ± 20 ms to 3 ± 1 ms at pH 9.5 (Supplementary Figs. 6a, 7, and 8a and Supplementary Table 3). Asp31 was intended to make a

secondary hydrogen bond to His71, analogous to the D1-Asn298/D1-His190/D1-Tyr161 ($Y_Z$) hydrogen bond network in PSII[49,50]. Without a crystal structure of the L31D/L71H RC maquette mutant, however, we cannot confirm that the Asp31-His71 hydrogen bond forms; it remains possible that Asp31 raises the quantum yield of electron transfer by altering the dynamics and hydration of Tyr168 rather than by acting directly as a proton acceptor for His71.

The carboxylate-rich binuclear metal binding site of the RC maquette extends the electron donor chain. Spectroscopic monitoring of Co(II) binding yields $K_D$ values of 0.5 and 15 μM for the first and second metal bound (Supplementary Fig. 3a). The Irving-Williams series[51] predicts the trend of metal complex stability as Mn(II) < Fe(II) < Co(II); thus the Co(II) affinity represents an approximate upper limit for Mn(II) and Fe(II) affinities.

Mn(II) binding stabilizes the binuclear metal site, increasing the RC maquette melting temperature (monitored by circular dichroism) from 81 °C to >100 °C. Binding of two Mn(II) ions per protein is confirmed by a bathochromic shift in the porphyrin Soret band of the ZnP/heme RC maquette (Supplementary Fig. 3).

Fe(II) binding is apparent from a similar Soret band shift. Bacterioferritin-like ferroxidase activity demonstrates that the diiron site is redox active (Supplementary Fig. 12). Light-induced electron transfer is clear as near-stoichiometric Fe(II) prolongs the lifetime of charge separation from 2.0 ms to 250 ms at pH 7.5 in RC maquette variant L71H (Fig. 3e, orange and Supplementary Figs. 8b and 10a). Tyr promotes electron transfer from Fe(II) to ZnP⁺ (Supplementary

Fig. 10), as knocking out Tyr168 (Y168L) lowers the yield of Fe(II)-facilitated charge separation to an undetectable level.

Unbound Fe(II) is not a viable electron donor at near-stoichiometric concentrations, or even at 10-fold excess, as seen by the suppression of Fe(II)-facilitated charge separation by pre-loading the maquette with stoichiometric Zn(II). When a very large excess of Fe(II) is added (1.1 mM) in the absence of Zn, Cd, or Mn, the kinetic behavior is altogether different from stoichiometric Fe(II) concentrations: the measured ferrous heme lifetime is greater than 10 s. Thus, at very high concentrations, Fe(II) can act as an exogenous electron donor and diffuse away before charge recombination.

At neutral pH, Zn(II) and Cd(II) suppress Tyr oxidation. These metals may electrostatically inhibit proton transfer from Tyr to protonatable residues in the metal cluster site. A similar effect could apply to redox-active bound Fe(II); however, even if transient Tyr oxidation were energetically uphill, subsequent energetically favorable electron transfer from Fe(II) could kinetically stabilize charge separation. In comparison, Mn(II) oxidation after transient Tyr oxidation will not be as energetically favorable, so less kinetic stabilization is expected.

Mn oxidation is not simple. During photo-assembly in natural PSII[46], initial Mn(II) photo-oxidation steps are coupled to a dark rearrangement that takes ~150 ms, while in modified purple bacterial reaction centers somewhat similar to our system, Mn(II) oxidation has a lifetime of 12 ms[22,26]. These rates are much slower than the rate of direct electron tunneling at these distances. We expect Mn(II) photo-oxidation to be slowed by site rearrangements on a timescale longer than the 2.7 ms charge recombination of the tyrosine-ZnP-heme triad (see Supplementary Figs. 8b and 9). Slower charge recombination is required for higher yields of Mn oxidation.

Like Fe(II), ferrocene (Fc) electron donors permit long charge separation lifetimes in the RC maquette (Fig. 1d), because their relatively low $E_m$ values inhibit the uphill reverse electron transfer from P to D$^+$. Maleimide-functionalized Fc ($E_m$ 0.56 V in DMF) is coupled to a Cys residue to make Fc168 or Fc164 (sequences are given in Supplementary Table 1). Fc $E_m$ values are modulated by the protein environment; Fc-modified azurin ranges from 0.51 to 0.55 V depending on surrounding amino acids[52]. Figure 3e (cyan) and Supplementary Fig. 5a give a 350 ms lifetime for charge separation in the Fc168-ZnP-heme B triad based on the heme redox signal.

At the electron accepting end of the RC maquette, we confirm the prediction (Fig. 1c) that the electron transport chain remains functional over a wide range of acceptor midpoint potentials. Replacement of heme B in the Fc168-ZnP-heme B triad with the Fe porphyrin DADPIX, which is 200 mV more oxidizing, changes the lifetime of charge separation from 350 ms to 20 ms and the yield of D$^+$PA$^-$ from 4% to 31% (see Supplementary Fig. 5).

As an acceptor, Co(III)PPIX achieves highly desirable electrochemical and photochemical proton reduction to H$_2$ in natural proteins[17]. It binds tighter to the RC maquette ($K_D$ 250 nM) than to the natural 4-helix bundle cytochrome $b_{562}$[17]. While bound CoPPIX is indeed photoreduced by the Zn porphyrin in the RC maquette under steady state illumination in the presence of ascorbate and EDTA as sacrificial electron donors, the quantum yield of long-lived charge separation using CoPPIX as the direct acceptor is low (see Supplementary Fig. 11). Work with CoPPIX in myoglobin ($E_m$ + 0.1 V) is orders of magnitude slower than heme reduction[53], a sign that additional reorganization barriers, such as ligand exchange, accompany CoPPIX reduction. Just as slow Mn photo-oxidation is likely to be assisted by extending the donor redox chain, designs using CoPPIX as an effective terminal reductive catalytic center must pair it with a rapid initial electron acceptor in a reducing electron transfer chain.

## Discussion
By exploiting the multi-cofactor adaptability of maquette design with permutations of donors and acceptors, we show how electron tunneling theory can guide the engineering of artificial proteins to harness light for directed electron transfer reactions. Observed charge separated states persist for hundreds of milliseconds, making the RC maquette a viable framework for photosynthetic redox catalysis. In contrast, chemically synthesized photochemical triads usually have charge separation lifetimes too short for chemistry; for example, the D$^+$PA$^-$ lifetime of a THF-solubilized Zn porphyrin-freebase porphyrin-C$_{60}$ compound was reported to be just 34 μs, a relatively long lifetime for a molecular triad[27]. Unlike synthetic constructs, artificial reaction center proteins co-opt cellular machinery to economically construct a nanometer scale scaffold. This synthetic biology approach takes advantage of the chemical functionality of natural amino acids while building on our understanding of the distance and energetic requirements for biological electron transport.

With the achievement of adequate charge separation lifetimes, the stage is now set for the challenging task of engineering catalytic sites, first by clarifying the principles for photo-oxidative assembly of Mn clusters in a framework that is simple, stable, and adaptable compared to PSII. The maquette method offers the promise of integrating artificial reaction centers with cellular light-harvesting systems in photosynthetic microbes or plants to divert energy into biofuel production, to couple water oxidation to fuel production in a single reaction center with high thermodynamic efficiency, or to create new arrangements and combinations of photosystems to improve solar input for photosynthesis.

## Methods
### Protein expression and purification
A gene encoding the RC maquette including an N-terminal His$_6$-tag and TEV protease cleavage sequence of MGKGGHHHHHHGGDGENLYFQ was purchased from DNA2.0 in a pJexpress414 vector with codons optimized for expression in *E. coli*. The TEV protease cleavage site is between Q21 and G22; G22 becomes G1 in the protein used for experimental studies discussed here. Primers for mutagenesis were purchased from Invitrogen, and mutant plasmids were PCR amplified using AccuPrime™ Pfx SuperMix (Invitrogen). Mutations including Y168L, Y168C, G164C, L71H, L31D, H124M, and E34A were prepared in this way. Full sequences of all RC maquette variants are given in Supplementary Table 1. The plasmids were transformed into BL21-(DE3) competent *E. coli* cells (New England Biolabs), and bacteria were grown to 0.6 OD at 600 nm. Expression was induced with between 40 μM and 1 mM IPTG for 4 h at 37 °C. Bacteria were pelleted by centrifugation, resuspended in 20 mM sodium phosphate buffer at pH 7.4 with 500 mM NaCl, 40 mM imidazole, 1% w/v octylthioglucoside, ~0.1 mg/mL DNAse, and lysed by homogenization or sonication. Lysates were centrifuged at 25,000 g for 30 min, and supernatants were applied to a 5 mL Ni-NTA HisTrap FF prepacked column (GE Healthcare Life Sciences) on an Akta FPLC. Protein was eluted in 250 mM imidazole, 500 mM NaCl, and 20 mM sodium phosphate at pH 7.4. The protein was treated with TEV protease and incubated at 4 °C for up to 2 weeks, applied to an Ni-NTA column again, and finally purified by size exclusion chromatography (SEC) using an XK 16/70 column (GE Healthcare Life Sciences) packed with 110 mL of Superdex 75 prep grade gel filtration medium (GE Healthcare Life Sciences). Product molecular weight was verified by Matrix Assisted Laser Desorption Ionization-Time of Flight mass spectrometry (MALDI-TOF) with a synapic acid matrix.

### Redox cofactors
Zn 5-phenyl 15-(p-carboxyphenyl) porphyrin (ZnP) was kindly provided by Dr. Tatiana Esipova from the research group of Sergei Vinogradov at the University of Pennsylvania. The synthesis details are described in Kodali et al.[12]. N-ferrocenyl maleimide lyophilized powder was purchased from Sigma-Aldrich. The two cysteine-containing RC maquette mutants (Y168C and Y168L/G164C) were subjected to the following

standard maleimide-cysteine reaction protocol: N-ferrocenyl mal-eimide was dissolved in DMF at 10 mM concentration. Appropriate dilution was made so that 50 μM of the maquette in 6 M Guanidine hydrochloride and 5 mM TCEP reacted with the N-ferrocenyl-maleimide at 1:10 ratio. The reaction was allowed to proceed with constant rotation overnight with protection from light. Labeled and unlabeled protein were separated by high-performance liquid chro-matography (HPLC) using a Vydac 218TP54 C18 column with a gradient of water/acetonitrile (ACN) mixture going from 35% ACN:65% water to 65% ACN:35% water over 40 min. The unlabeled protein eluted at 51% ACN:49% water while the labeled protein eluted later at 55% ACN:45% water. The purity of the labeled protein fraction was examined and verified using MALDI-TOF.

## Ultraviolet/visible spectroscopy

Ultraviolet/visible spectroscopy (UV/vis) was performed using a Varian Cary-50 spectrophotometer at room temperature. RC maquette con-centration was determined using an extinction coefficient of $\varepsilon_{280nm} = 12490$ $M^{-1}cm^{-1}$ attributed to Tyr and Trp absorbance. Heme B stock concentrations were measured by hemochrome assay[54]. Stock con-centrations of ZnP and metal salts were estimated by mass measure-ment of dry powder. Tetrapyrrole stock solutions were prepared in dimethyl sulfoxide (DMSO).

## Spectro-electrochemistry

UV/vis light from an Ocean Optics light source was guided through fiber optic through a fine gold-honeycomb working electrode (Pine Instruments) surface-modified with cystamine and passed via fiber optics to an Ocean Optics spectrometer. The gold working and counter electrodes relative to an Ag/AgCl reference electrode, were set by a CH Instruments Electrochemical Analyzer. The following redox mediators[55] were used to speed redox equilibration to a few minutes at each potential: for DADPIX and B-heme, the redox mediator mix was anthra-quinone-2-sulfonate (100 μM), benzyl viologen (50 μM), methyl viologen (50 μM), indigo trisulfonate (50 μM), phenazine (50 μM). ZnP titration used a mixture of high potential mediators $K_2[IrCl_6]$ (20 μM), 2,6-p-quinone (20 μM), $K_4[Mo(CN)_8]$ (20 μM), dicyano bis(1,10-phe-nanthroline) iron(II) dehydrate (10 μM), $K_3[Fe(CN)_6]$ (20 μM).

## Circular dichroism

Circular dichroism measurements were made using an Aviv Model 410 instrument. Far UV CD spectra were measured from 190 to 260 nm with a 1 nm bandwidth every 1 nm for 5 s per data point and were performed in a 0.1 cm path length quartz cuvette. Each reported spectrum was the average of three spectra that were measured con-secutively. Thermal denaturation was measured at 222 nm for 5 s with a 2 nm bandwidth, and data were collected every 2 °C with a heating rate of 10 °C/min followed by a 4 min incubation time.

## Crystallography

All RC maquette crystals were grown in hanging drops at 4 °C. Protein stock solutions contained 20−40 mM NaCl and 10−20 mM piperazine-N,N'-bis(2-ethanesulfonic acid) (PIPES) buffer at pH 6.5 and were stored at 4 °C. Small amounts of porphyrin cofactors were added at room temperature from dimethyl sulfoxide stock solutions until a 1.0:1.0 equimolar ratio of porphyrin to protein was reached. Several minutes of equilibration time were given between additions of porphyrin in order to prevent aggregation and precipitation of cofactor. (Heme B was added prior to ZnP for crystal structure in PDB ID: 5VJS). Sample buffers were exchanged to the low salt PIPES buffer using 5 kDa MWCO Vivaspin Turbo 15 concentrators (Sartorius AG). Supplementary Table 4 gives for each crystal the RC maquette stock concentration used, cofactors included in the stock solution, well solution composition, drop volume, and cryoprotectant for each crystal structure. Crystal structure 5VJU used streak seeding with a cat whisker.

All X-ray crystallographic data sets were collected from single crystals at 100 K. X-ray diffraction data for crystal structures with PDB IDs 5VJS and 5VJT were collected at the National Synchrotron Light Source, beamline X6A using an ADSC Q270 CCD x-ray area detector. X-ray diffraction data for the crystal structure of L71H mutant (PDB ID: 5VJU) was collected using a Rigaku Micromax-007 HF rotating copper anode X-ray generator and VariMax HF optics with a Rigaku Saturn 944 HG CCD detector. Multiwavelength anomalous dispersion (MAD) data was collected to solve crystal structure 5VJS. Data were integrated with XDS[56] and initial phases were calculated using SOLVE[57]. All other crystals used the 5VJS crystal structure as a model for molecular replacement using Phaser[58]. Software packages CCP4i[59] and PHENIX[60,61] were used throughout the structure solution and refine-ment process. Intensities were scaled using SCALA[62]. Refinement was done using REFMAC5[63], phenix.refine[64], and PDB_REDO[65]. Real space refinement was done by manually fitting models into electron density maps in Coot[66]. Density modification was done using RESOLVE[67]. Composite omit maps were created with simulated annealing in PHE-NIX to reduce model bias[64,68,69]. Omit maps of cofactor binding sites are presented in Supplementary Fig. 2. Polder omit maps in Supple-mentary Fig. 13 were also created in PHENIX[70]. Statistics for the three data sets collected for MAD solution of crystal structure 5VJS are given in Supplementary Table 5. Data collection and refinement statistics for all three crystal structures, 5VJS, 5VJT, and 5VJU, are given in Supple-mentary Table 6. Protein visualization and preparation of X-ray crystal structure images for figures was done in The PyMOL Molecular Gra-phics System, Version 2.5.0a0, Schrödinger, LLC.

## Transient absorption

Samples contained ~3.5 μM protein and 50 mM NaCl. For samples at pH 7.5 the buffer was 10 mM 3-(N-morpholino)propanesulfonic acid (MOPS), and at pH 9.5 the buffer was 10 mM N-cyclohexyl-2-aminoethanesulfonic acid (CHES). Where the electron acceptors heme B or DADPIX were used, the acceptor was added to the protein solution prior to ZnP from a DMSO stock solution. We used 1.0 molar equivalent of acceptor and ~0.8 equivalents of ZnP per pro-tein. Solutions containing $FeCl_2$, $MnCl_2$, $CdCl_2$, or $ZnSO_4$ used a 10 μM concentration of the metal salt. Protein solutions were maintained anaerobic under argon scrubbed by a reducing $VSO_4$ solution[71].

Transient absorption experiments were performed at 20 °C with a 10 Hz Q-switched frequency-doubled Nd:YAG laser (532 nm, DCR-11 Spectra Physics). A Xe flash lamp emitted probe beams of white light into split fiber optic cables with a gate width of 600 ns to 1 μs at delay times after the laser flash to measure the change in absorbance spectra as a function of time after the laser flash. The split fiber optic cables were used to direct one probe beam through the laser-pumped (experimental) region and the other probe beam through the dark (reference) region of the sample in the cuvette. Each probe beam was collected by a fiber optic bundle and flattened into a line of fibers at the entrance slit of an Acton SP-2156 spectrograph. The images from the exit slit of the monochromator were focused onto a Princeton Instruments PiMax-3 ICCD camera operated with WinSpec32 proprie-tary software. Delay times were set by a Stanford Research System DG 535 digital delay generator. A ThorLabs beam shutter allowed the 10-Hz laser repetition rate to be lowered to seconds for measurements at times longer than 100 ms.

Difference spectra were collected for delay times of 1 μs to ~1 s in intervals of half log time units. The same sample was measured at different delay times, and at least 50 spectra were collected for each sample at each delay time. Comparison of difference spectra at 1 μs that were collected at the beginnings and ends of transient absorption experiments revealed no significant photodegradation over the course of the experiments. Experiments were repeated using independently-prepared samples as indicated in Supplementary Table 3 to determine

standard deviations in electron transfer rate and charge separation quantum yield measurements.

## SVD analysis

Singular value decomposition (SVD) is applied to the data matrix obtained from preprocessing of the raw transient absorption data. The analysis in Mathematica (Wolfram) calls the built-in "svd" function on the raw data. The raw data is represented by matrix A, with dimensions l by t, where l is the number of wavelengths and t is the number of time-resolved snapshots taken during the experiment. Calling the "svd" function using A as the input variable returns a left matrix U with dimensions l by l, a central matrix S with dimensions l by t, and a right matrix V with dimensions t by t.

The right matrix V resulting from the SVD is used to fit a kinetics model described by solutions of ODEs depending on the configuration of the photosystem examined. Because the redox states of the cofactors differ by more than 60 mV, SVD spectra resemble pure charge-separated states, rather than an equilibrium between redox states. The model fitting follows a modified version of the method described by Henry and Hofrichter[72] where a linear transformation matrix C is applied to the expected population matrix P computed from the model. The $L_2$ norm of the difference between the right t by t matrix and the product CP is minimized by the standard minimization function, in which the rates of various electron transfers in the kinetics model along with the elements of the C matrix are allowed to vary. The minimization function is instructed to utilize the quasi-newton minimization procedure. Various initial conditions are tried in order to obtain the best fitting of the data. The result of the minimization is examined by comparing the residual of the minimization to other values obtained from different sets of initial conditions to ensure that the minimum discovered by the algorithm is, as far as the information reveals, the global minimum or a satisfying local minimum. The values of the C matrix as well as the rates of the electron transfer between the states are then used to compute the reconstructed elemental spectra that make up the observed raw data as well as the electron transfer kinetics that best explains the observed data.

## Electron-tunneling yield calculations

The contour plot of Fig. 1c and d are calculated by solving the set of differential equations for the kinetic scheme of Fig. 1b, beginning with the triplet state $D^3P^*A$ at time zero, and evolving to the states $DP^*A^-$, $D^+PA^-$ and DPA. We use the electron tunneling expression Eq. (1) to predict exergonic electron transfer reaction rates:

$$\log(k_{ET}) = 13 - 0.6(R - 3.6) - 3.1(\Delta G + \lambda)^2/\lambda \qquad (1)$$

The electron transfer rate $k_{ET}$ is in units of $s^{-1}$, the edge-to-edge distance $R$ is in Å, the driving force between donor and acceptor $\Delta G$ (usually estimated from the difference between their redox midpoint potentials, $E_m$) is in eV, and the Marcus reorganization energy $\lambda$ is in eV. For electron transfers slower than 20 ps, the reorganization energy ranges from 0.4 for large redox cofactors (such as tetrapyrroles) buried in non-polar protein environments, to 1.4 eV for more compact redox centers or polar environments[1,3,29]. When $\lambda$ has not been measured, a middling value around 0.8 eV will typically provide a rate estimate within an order of magnitude. Uphill reverse electron transfer rates can be estimated by using the downhill rate and applying a Boltzmann thermodynamic penalty of 1 order of magnitude of rate for every 0.06 eV uphill[1,3,29,73].

The non-electron-tunneling rate of phosphorescence decay of the triplet state is 250 s$^{-1}$. The set of differential equations are solved in Mathematica for the fractional yield of the charge separated $D^+PA^-$ state 100 µs after time zero for plotting. For Fig. 1c, the edge-to-edge distance between P and A is varied, as is the $E_m$ of the acceptor A.

Other tunneling parameters are held constant: D-P distance 4.6 Å, D-A distance 23.0 Å, $E_m$ D 0.72 V corresponding to tyrosinate donor, $E_m$ P 0.91 V, triplet photon energy 1.6 eV, reorganization energy 0.8 eV. For Fig. 1d, the edge-to-edge distance between P and D is varied, as is the $E_m$ of the donor D, while the P-A distance is held constant at 13.2 Å and the $E_m$ of A set at −0.19 V, corresponding to heme B acceptor.

## Reporting summary

Further information on research design is available in the Nature Research Reporting Summary linked to this article.

## Data availability

X-ray crystallographic coordinates and data files were deposited at the Protein Data Bank (PDB) with accession codes 5VJS (RC maquette with heme B, ZnP, and Zn2+), 5VJT (RC maquette with heme B and Zn2+), and 5VJU (RC maquette L71H mutant with heme B and Cd2+). Source data are provided with this paper.

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

## Acknowledgements

We thank T.V. Esipova and S.A. Vinogradov for generously providing the synthetic porphyrin ZnP used in this work. Funding was provided by the Department of Energy, Office of Science, Office of Basic Energy Sciences, through the Photosynthetic Antenna Research Center (PARC), an Energy Frontier Research Center, under Award DESC0001035.

## Author contributions

N.M.E., Z.Z., P.L.D., and C.C.M. conceived the project. B.M.D. contributed to the conception and design of experiments. N.M.E. designed and crystallized the RC maquette protein. S.E.S. and N.M.E. collected and processed X-ray diffraction data and solved crystal structures. N.M.E. and Z.Z. purified proteins. Z.Z. crosslinked N-ferrocenyl maleimide to cysteine-containing RC maquette mutants. N.M.E., Z.Z., and C.C.M. conducted transient absorption experiments and analyzed data. N.M.E. and C.C.M wrote the manuscript with contributions from Z.Z., S.E.S., B.M.D. and P.L.D.

## Competing interests

The authors declare no competing interests.
