## [Peer Review File · Nature Communications]

Reviewers' Comments:

Reviewer #2:

Remarks to the Author:

The authors have adequately addressed my concern regarding describing the design approach and discussing design factors associated with using a helical scaffold. It is an exciting project in scope and the manuscript is clear and convincing.

Reviewer #3:

Remarks to the Author:

I read through the revised manuscript and response letter. The changes are only concerning the wording, there are no new experiments as requested by myself and the other reviewers. Therefore, my original concerns still stand and I decline to support publication of this paper.

Second report from Reviewer #3 (received April 26, 2022):

“I read through the revised manuscript and response letter. The changes are only concerning the wording, there are no new experiments as requested by myself and the other reviewers. Therefore, my original concerns still stand and I decline to support publication of this paper.”

Response from the authors:

We would like to thank Reviewer #3 for taking the time to evaluate our manuscript and revisions. We wish to stress that all of the major findings of this paper are well-supported by the experimental evidence. We provide high resolution X-ray crystal structures and spectroscopic data to prove that we have designed a four helix bundle protein that assembles electron donors, a pigment, and electron acceptors. One of the crystal structures is the first experimental structure of a de novo designed protein that binds two unique porphyrins with site specificity. Referring to our crystallographic data, Reviewer #3 states, “... *these data provide very strong support for the design of the protein*” In addition, we present transient absorption data that includes

direct spectroscopic evidence that heme B and DADPIX electron acceptors are photoreduced in the RC maquette. We have shown that these electron acceptors stay reduced for as long as hundreds of milliseconds and that the kinetics of charge recombination and the quantum yield of charge separation depend upon the nature of the electron donor. Thus, we have provided clear evidence that a functional DPA charge separating triad has been created using a de novo designed protein.

As we explained in our previous rebuttal, Reviewer #3 made several incorrect statements in his/her first report (reproduced above), which all stem from the same factual misunderstanding of how we calculated charge separation lifetimes and quantum yields in transient absorption experiments. These statements include the following:

1. *“... There is no direct evidence of electron transfer, it is all inferred from changes in the decay rate of the Zn-porphyrin.”*
2. *“There is no direct evidence that the heme serves as an electron acceptor”*
3. *“In general, their arguments for multiple electron transfer processes are based upon around [sic] measurements of the decay rate of the Zn-porphyrin after excitation.”*

It is difficult to understand why this reviewer believes there is no evidence of heme acting as an electron acceptor. We demonstrate in several constructs the unambiguous light-induced appearance and decay of the heme redox spectrum, and as a control, the absence of a heme redox spectrum when the heme is chemically pre-reduced. Further, we demonstrate the unambiguous light induced appearance and decay of the DADPIX redox spectrum when heme is replaced by DADPIX. These characteristic spectral signatures strike us as the clearest possible evidence for the direct participation of a redox center in light-activated electron transfer. Heme and DADPIX redox signals from transient absorption experiments can be found in Fig. 3 and Ext. Data Figs. 25 and 7.

Our conclusions surrounding electron transfer rates do not depend solely upon observations of Zn porphyrin absorbance features, as Reviewer #3 suggests. Rather, rates of electron transfer and quantum yield measurements were calculated using spectroscopic signals from both the pigment (Zn porphyrin) and the electron acceptor (heme B or DADPIX). The heme and DADPIX redox signals give us the information needed to calculate charge separation lifetimes and quantum yields. Indeed, it would not be possible to estimate charge separation lifetimes if electron transfer were inferred from changes in the decay rate of the Zn porphyrin alone, as the Zn porphyrin is spectroscopically silent in the transition from the D^+PA^- charge separated state back to the DPA ground state.

Reviewer #3's concerns regarding heme photoreduction were addressed by the addition of a reference heme redox spectrum to Figure 3d in which an absorbance spectrum of air-oxidized RC maquette-heme complex was subtracted from a spectrum of the dithionite-reduced species in the dark. The agreement between the reference heme redox spectrum (black dotted trace) and

the transient absorption signal that outlives the Zn porphyrin bleach recovery (blue trace) clearly and unambiguously shows that heme serves as the electron acceptor. We also clarified the language in the caption of Ext. Data Fig. 1, which Reviewer #3 misunderstood as signifying that we did not detect heme reduction in any transient absorption experiment. In fact, Ext. Data Fig. 1 further validates our claim that an electron donor is necessary for long-lived charge separation. In addition, we added Supplementary Table 2 to provide error analysis of transient absorption experiments in which heme reduction was observed directly. In summation, we have addressed possible sources of the misunderstanding to make clear that we present direct evidence of heme and DADPIX reduction and that spectroscopic signals from both the pigment and the acceptor support our conclusions.

“*There is no direct evidence that the designed tyrosine is redox active ...*” Reviewer #3’s skepticism in this regard is easier to understand. Tyrosine does not have as dramatic a light induced redox spectral change as the colorful tetrapyrroles of heme, DADPIX and ZnP, so its spectral signature is not expected to be directly obvious in the time resolved measurements. EPR measurements are used in some (but not all) published reports as evidence for light-activated tyrosine oxidation in model systems. However, identifying tyrosine by EPR alone is not a trivial experiment, as the spin signature can easily be obscured by other light induced radicals generated by the accessory redox reagents; these reagents are used to assure that the Heme/ZnP/Tyr redox centers are either poised in redox states that support full charge separation, or as a control, have heme reduced and thus cannot support light induced electron transfer. In any event, such EPR experiments cannot now be performed as both P.L. Dutton and C. C. Moser have retired from the University of Pennsylvania and closed the laboratory. For this reason, we rely on other methods, as used in other reports (DOI: doi.org/10.1002/cptc.202100014; PMID: PMC4195373), to determine if tyrosine is likely to be active in the charge transfer relay. This includes, most importantly, the effect of knocking out the tyrosine by mutagenesis, and also modulating the energetics of tyrosine oxidation by protonation/deprotonation as a function of pH. These straightforward effects will persuade many, if not all, readers of the likely participation of tyrosine in the electron transfer relay. While evidence of tyrosine oxidation adds to the interest of our report, there are many other firsts established in this de novo design of reaction centers that represent important successes that will be of interest to many readers.

Our manuscript shows how the charge separation lifetime and quantum yield are influenced by different electron donors including ferrocene, Fe(II), and a tyrosine amino acid. When no electron donor is included, a heme redox signal is not observed (Ext. Data Fig. 1), indicating that an electron donor of some kind is needed to form a long-lived charge separated state. This is a central conclusion of the paper, and it shows that we have successfully created DPA triads for charge separation. Reviewer #3 correctly pointed out that, “... *there are no direct experimental measurements of the redox properties of the metal cluster in either the dark or light-induced states.*” We reported the midpoint potential of ferrocene, but Figure 1d illustrates that electron donor midpoint potentials can vary across a wide range encompassing the likely potential of a diiron site with little effect on the quantum yield of charge separation. Nevertheless, we have added Ext. Data Fig. 10e to the present manuscript revision to provide direct evidence of redox activity in the diiron site. Ext. Data Fig. 10e presents previously unreported experimental data

concerning ferroxidase activity catalyzed by the RC maquette. We observe an increase in absorbance in the ultraviolet range attributed to oxidation of Fe(II) to Fe(III) upon exposure to oxygen. The Fe(III) extinction coefficient in the RC maquette at 300 nm is in good agreement with extinction coefficients measured in previous natural and designed diiron four-helix bundle proteins (PMCID: PMC2374189; PMID: 10769150). This gives direct spectroscopic evidence that the iron center is redox-active and has redox properties similar to those of other well-studied diiron helical bundle proteins.

Regarding our last round of revisions, Reviewer #3 said in the second report, “*The changes are only concerning the wording, there are no new experiments as requested by myself and the other reviewers.*” In fact, we added several new sets of data to the revised manuscript. We included previously unreported transient absorption experiments in order to provide error analysis in Supplementary Table 2. These data lend additional support to our conclusions surrounding light-driven charge separation. In addition, we presented details of metal binding thermodynamics in Extended Data Figure 10. The new polder omit maps in Supplementary Figure 2 were added as a direct response to Reviewer #3’s concerns regarding cofactor electron density maps in the X-ray crystal structures. These revisions go significantly beyond simple changes in the wording.

We have addressed Reviewer #3’s concerns by the addition of several sets of experimental data including direct evidence of redox activity in the metal site (Ext. Data Fig. 10e), error analysis on previously unreported transient absorption data that provide more evidence of long-lived charge separation (Suppl. Table 2), polder omit maps that verify the presence of ligands in crystal structures (Suppl. Fig. 2), and detailed data on metal binding thermodynamics (Ext. Data Fig. 10a-d). Together, these changes have strengthened the manuscript, making it deserving of publication in Nature Communications.

Reviewers' Comments:

Reviewer #2:

Remarks to the Author:

no further comments to the authors